# Antioxidant and Neuroprotective Properties of Selected Pyrrole-Containing Azomethine Compounds in Neurotoxicity Models In Vitro

**DOI:** 10.3390/ijms26093957

**Published:** 2025-04-22

**Authors:** Denitsa Stefanova, Alime Garip, Emilio Mateev, Magdalena Kondeva-Burdina, Yordan Yordanov, Diana Tzankova, Alexandrina Mateeva, Iva Valkova, Maya Georgieva, Alexander Zlatkov, Virginia Tzankova

**Affiliations:** 1Department Pharmacology, Pharmacotherapy and Toxicology, Faculty of Pharmacy, Medical University-Sofia, 1431 Sofia, Bulgaria; alime_1999@abv.bg (A.G.); magdalenakondeva@gmail.com (M.K.-B.); yyordanov@pharmfac.mu-sofia.bg (Y.Y.); vtzankova@pharmfac.mu-sofia.bg (V.T.); 2Department “Pharmaceutical Chemistry”, Faculty of Pharmacy, Medical University-Sofia, 1431 Sofia, Bulgaria; e.mateev@pharmfac.mu-sofia.bg (E.M.); d.tsankova@pharmfac.mu-sofia.bg (D.T.); a.dineva@pharmfac.mu-sofia.bg (A.M.); mgeorgieva@pharmfac.mu-sofia.bg (M.G.); azlatkov@pharmfac.mu-sofia.bg (A.Z.); 3Department “Chemistry”, Faculty of Pharmacy, Medical University-Sofia, 1431 Sofia, Bulgaria; ivalkova@pharmfac.mu-sofia.bg

**Keywords:** pyrrole-based compounds, in vitro neuroprotection, brain subcellular fractions, SH-SY5Y, in silico, ADME

## Abstract

Neurodegenerative diseases involve oxidative stress and enzyme dysregulation, necessitating novel neuroprotective agents. This study evaluates the neuroprotective and antioxidant potential of seven pyrrole-based compounds with predicted radical scavenging activity and inhibitory effects on monoamine oxidase B (MAO-B) and acetylcholinesterase (AChE). The compounds were tested in vitro using SH-SY5Y neuroblastoma cells and subcellular rat brain fractions, including synaptosomes, mitochondria, and microsomes. Neuroprotective and antioxidant effects were assessed in oxidative stress models, including H_2_O_2_-induced stress in SH-SY5Y cells, 6-hydroxydopamine toxicity in synaptosomes, tert-butyl hydroperoxide-induced stress in mitochondria, and non-enzyme lipid peroxidation in microsomes. In silico screening for lipophilicity, hydrogen bonding, total polar surface area (TPSA), and ionization properties, was performed to evaluate bioavailability. All compounds exhibited a weak neurotoxic effect on the subcellular fractions at a concentration of 100 µM. However, in oxidative stress models, they demonstrated significant neuroprotective and antioxidant effects at 100 µM. In SH-SY5Y cells, compounds **7**, **9**, **12**, **14**, and **15** exhibited low toxicity and strong protective effects at concentrations as low as 1 µM. In silico analysis prioritized compounds **1**, **7**, **9**, **12**, and **14** for further development based on their favorable bioavailability. The tested pyrrole-based compounds exhibit promising neuroprotective and antioxidant properties, with several candidates showing potential for further development based on both in vitro efficacy and predicted oral bioavailability.

## 1. Introduction

Neurodegenerative disorders have a complex pathogenesis, influenced by both genetic and environmental factors. Several hypotheses aim to explain their underlying mechanisms, such as oxidative stress, mitochondrial dysfunction, and neuroinflammation. Oxidative stress, a key factor in neurodegeneration, results from an imbalance between reactive oxygen species (ROS) and the body’s antioxidant defenses. Post-mortem brain analyses have shown increased markers of lipid peroxidation, protein carbonylation, and DNA/RNA oxidation products, such as 8-hydroxydeoxyguanosine. Additionally, animal models using substances like 1-methyl-4-phenyl-1,2,3,6-tetrahydropyridine (MPTP), rotenone, paraquat, and 6-hydroxydopamine (6-OHDA) provide further evidence supporting this theory [1]. ROS, mainly superoxide anions, hydrogen peroxide, and hydroxyl radicals, drive neurodegeneration in the context of reduced glutathione levels (GSH) and high iron and calcium contents in the substantia nigra. Mitochondrial dysfunction and the resulting oxidative stress are key characteristics of neurodegeneration. Microglia-triggered neuroinflammation in the substantia nigra exacerbates neurodegeneration, as shown by post-mortem studies and PET evidence (positron emission tomography) [2]. Targeting ROS in neurodegenerative diseases could slow their progression and improve patient outcomes [3]. Oxidative stress is frequently linked to neurodegenerative diseases due to the brain’s high oxygen consumption and limited antioxidant defenses. It can contribute to neuronal death through mechanisms such as mitochondrial dysfunction, disrupted proteostasis, neurotransmitter metabolism disturbances, inflammation, or compromised antioxidant pathways. Elevated levels of reactive oxygen species (ROS) and reactive nitrogen species (RNS), combined with impaired antioxidant defenses, are commonly observed in disorders like Alzheimer’s, Parkinson’s, Huntington’s, and amyotrophic lateral sclerosis. Research using animal and cellular models indicates that ROS and RNS play a crucial role in damaging cellular components, ultimately leading to cell death. Additionally, genetic mutations in antioxidant defenses further emphasize the brain’s heightened susceptibility to oxidative stress in these diseases [4].

Recently, there has been particular interest in the role of pyrrole-containing compounds in neurobiology. A novel group of pyrrole-containing compounds is characterized by an azomethine residue (–NH–N=CH–), linked to a carbonyl group [5]. Compounds with such functional groups exhibit a broad spectrum of pharmacological effects. These effects include antimicrobial [5], antitumor [6], anticonvulsant [7], antituberculosis [6], anti-inflammatory [8], analgesic [9], and antioxidant activities [10,11], among others. The diverse therapeutic potential of these compounds has been well-documented in the literature, highlighting their significance in various areas of medical research and drug development [12]. There are also approved drug substances with an azomethine residue attached to a carbonyl group, such as the antibacterial agents nitrofurazone, furazolidone, and nitrofurantoin [13]. Current studies provide evidence for the high radical binding activity of compounds containing a hydrazide–hydrazone group. For example, Devi and Pachwania (2021) reported on the synthesis, characterization, and in vitro antioxidant and antimicrobial activities of diorganotin (IV) complexes derived from hydrazide Schiff base ligands [14]. Additionally, good AChE and MAO-B inhibitory activities of newly synthesized hydrazide–hydrazones have been discussed by Coşar et al. (2022), who investigated the anticholinesterase activities of novel indole-based hydrazide–hydrazone derivatives, including their design, synthesis, biological evaluation, molecular docking studies, and in silico ADME predictions [15].

P. Rawat et al. (2020) synthesized and characterized pyrrole–chalcone compounds using both experimental techniques and theoretical calculations [10]. The study revealed that these compounds show promising free radical scavenging properties and Fe^2+^ ion chelation activity. Through detailed structural analysis and evaluation of their antioxidant properties, the research highlights the potential of pyrrole–chalcone derivatives in alleviating oxidative stress. These findings suggest that these compounds could be further investigated for their therapeutic potential, particularly in treating conditions related to oxidative damage and metal ion-induced toxicity [10]. In another study, a new conjugate compound, pyrrolylcarnosine, was synthesized by combining carnosine with the aromatic five-membered nitrogen heterocycle pyrrole. The findings revealed that pyrrolylcarnosine exhibits high antioxidant activity in experimental models. It also demonstrated a neuroprotective effect against oxidative stress induced by the neurotoxin 2,2′-azobis (2-methylpropionamide)-dihydrochloride (AAPH), increasing the viability of differentiated human neuroblastoma SH-SY5Y cells and protecting them from cell death. Overall, these results suggest that pyrrolylcarnosine holds promise as the basis for a new therapeutic drug [16].

The unsubstituted pyrroles and pyrrole derivatives containing an active hydrogen N-H atom in the core ring exhibit proven antioxidant activity, where the additional presence of substituents that accept or donate electrons in the ring may play an important role in its antioxidant activity [17,18]. This combined with the current studies determining the radical binding capacity of the hydrazide–hydrazone group pointed our aim towards combining these two active principles. Thus, we synthesized few pyrrole-based azomethines and evaluated their radical scavenging capacity [19,20]. The results of these evaluations determined the choice of targets for this study’s representatives.

Given the significant role of oxidative stress in neurodegenerative diseases, we aimed to investigate the antioxidant and neuroprotective effects of selected pyrrole-containing azomethine compounds with known radical scavenging activity, along with their MAO-B and AChE inhibitory effects. The compounds were tested across various oxidative stress models at both the subcellular and cellular levels to assess their impact on neurotoxicity. Our primary focus was to characterize their safety profiles and evaluate their potential protective effects in different in vitro models. Additionally, as a preliminary drug-likeness screening, the physicochemical properties of the compounds were calculated in silico.

The aim of this study is to evaluate the neuroprotective and antioxidant potential of pyrrole-based compounds, focusing on their radical scavenging activity and inhibitory effects on MAO-B and AChE, in various in vitro oxidative stress models. The study seeks to identify promising candidates for further development by assessing their safety profiles, neuroprotective effects, and bioavailability, with the ultimate goal of contributing to the treatment of neurodegenerative diseases associated with oxidative stress.

## 2. Results

### 2.1. Selection of the Model Compounds

For the current study, seven pyrrole-based compounds with previously reported MAO-B and AChE inhibitory capacities were selected. The target molecules were chosen based on their promising MAO-B (compound **11**) and AChE (compounds **9**, **11**, **14**, and **15**) inhibitory effects [21] and established radical scavenging properties (compounds **7**, **11**, and **12**) [22].

### 2.2. Synthesis of the Selected Molecules

For the purpose of this investigation, the selected compounds were resynthesized through the microwave (MW)-assisted approach by following the synthetic pathway presented on Figure 1 [21,22]. The synthesis generally follows a three-stage process representing initial esterification of the applied carboxylic acid (a) accompanied by hydrazinolysis (b) and pursued by condensation with the corresponding aldehyde (c) until formation of the final hydrazone molecule. The reactions were carried out through a conventional (conv.) and/or microwave-assisted (MW) approach (Figure 1).

### 2.3. In Silico Characterization of the Physicochemical Properties of the Target Compounds

Table 1 presents the molecular weights (MWs), pKa values, logP, logD7.4 (distribution coefficient at pH 7.4), topological polar surface areas (TPSAs), count of free rotatable bonds (RBs), hydrogen bond donors (HBDs), and hydrogen bond acceptors (HBAs) of the molecules, along with the count of the violations of Lipinski’s Rule of Five (ROF). This rule was applied for filtering out molecules with unfavorable properties and optimizing the experimental costs.

### 2.4. In Vitro Toxicity Evaluation of the Target Pyrrole Hydrazones on Subcellular and Cellular Models

#### 2.4.1. Effects on Isolated Rat Brain Synaptosomes

We assessed the impact of the compounds on biomarkers that characterize the functional–metabolic profile of brain synaptosomes in the absence of other prooxidant stimuli. At a concentration of 100 µM, the test substances demonstrated mild neurotoxic effects compared to the non-treated control synaptosomes. This was indicated by slight decreases in both synaptosomal viability and the levels of reduced glutathione (GSH) (Figure 1).

The performed experiments indicated that the investigated compounds decreased synaptosomal viability in the range of 22% to 29% compared to the control (*p* < 0.01).

#### 2.4.2. Effects on Isolated Rat Brain Mitochondria

We assessed the toxicity of the compounds using isolated brain mitochondria as another well-established subcellular model. Rat brain mitochondria were treated with the target compound at a concentration of 100 µM. The evaluated substances exhibited a mild, statistically significant neurotoxic effect compared to the controls (untreated mitochondria). This effect was characterized by a slight increase in malondialdehyde (MDA) production and a decrease in reduced glutathione (GSH) levels (Figure 2).

Notably, the tested compounds decreased the GSH levels by 34% to 39% compared to the control. In terms of MDA production, the evaluated compounds increased the MDA levels by 21% to 31% compared to the control.

#### 2.4.3. Effects on Isolated Rat Brain Microsomes

When administered at a concentration of 100 µM, the tested substances induced a weak prooxidant effect in isolated rat brain microsomes, as compared to the untreated control group. This effect was evaluated by measuring the levels of malondialdehyde (MDA), a commonly used biomarker indicative of lipid peroxidation and oxidative membrane damage. As shown in Figure 3, treatment with the compounds resulted in a modest increase in MDA production, reflecting a slight promotion of oxidative processes under the experimental conditions.

More specifically, several compounds caused a statistically significant rise in MDA levels relative to the control. Compound **1** increased MDA formation by approximately 33%, while compound **7** produced a 29% increase. Compound **9** led to a 38% elevation, and compound **11** exhibited the most pronounced effect, with a 41% rise. In addition, compound **12** increased the MDA levels by 35%, compound **14** by 39%, and compound **15** by 40%. These changes, although not extreme, consistently suggest a mild prooxidant activity across the tested series.

It is important to highlight that, while the observed elevations in the MDA levels indicate a shift toward oxidative processes, the magnitude of these changes remains relatively moderate. This suggests that the compounds did not cause severe lipid membrane damage at the tested concentration. Instead, they demonstrated a subtle prooxidant potential, which could reflect concentration- or context-dependent redox behavior commonly seen in bioactive compounds with electron-donating or -accepting functional groups.

Overall, these results underscore the necessity of further investigations into the redox properties of these molecules, particularly in environments with induced oxidative stress or at varying concentrations, to better understand their potential biological impact and therapeutic relevance.

#### 2.4.4. In Vitro Cytotoxicity Evaluation in Neuroblastoma Cell Line SH-SY5Y

To determine the in vitro safety of the selected compounds, we assessed their effects on SH-SY5Y cell viability. SH-SY5Y cells were treated with compounds **1**, **7**, **9**, **12**, **14**, and **15** at concentrations of 0.1, 1, 5, 10, 50, 100, 250, and 500 µM. The calculated IC_50_ values are shown in Table 2. All compounds exerted low toxicity, with IC_50_ values ranging from 53 to 263.5 µM. Compounds **7**, **9**, **12**, **14**, and **15** are the most promising candidates showing a good safety profile (IC_50_ values could not be determined within the concentration range of 0.1–500 µM). Compounds **11** and **1** showed low to moderate cytotoxicity on neuronal cells with calculated IC_50_ of 263.5 µM and 53.0 µM, respectively.

Following the determination of the in vitro cytotoxicity of the title compounds, we investigated their potential antioxidant and neuroprotective effects in various subcellular and cellular models of oxidative stress.

### 2.5. Protective Effects of the Target Compounds in In Vitro Models

#### 2.5.1. Effects on a Model of 6-OHDA-Induced Neurotoxicity in Isolated Rat Brain Synaptosomes

6-OHDA applied alone at a concentration of 150 µM to isolated synaptosomes caused a significant decrease by 55% in the synaptosomal viability and a reduction in the GSH levels by 50% compared to the control group (untreated synaptosomes) (Figure 4A,B). In contrast, all tested pyrrole hydrazones (100 µM) showed strong neuroprotective effects in the model of 6-OHDA-induced toxicity (Figure 4A,B). We found that compound **1** preserved the synaptosomal viability by 67%, compound **7** by 73%, compound **9** by 78%, compound **11** by 69%, compound **12** by 82%, compound **14** by 60%, and compound **15** by 64% compared to 6-OHDA alone. In terms of GSH levels, compounds **1**, **9**, and **11** preserved the levels by 60%; compounds **7**, **14**, and **15** by 70%; and **12** by 80% compared to the 6-OHDA group.

#### 2.5.2. Effect of Pyrrole Hydrazones on t-BuOOH-Induced Oxidative Stress in Brain Mitochondria

*t*-BuOOH alone caused a significant 209% increase in malondialdehyde (MDA) production and a 50% reduction in GSH levels in isolated brain mitochondria compared to control untreated mitochondria (Figure 5A,B).

When the test substances at a concentration of 100 µM were combined with *t*-BuOOH, all demonstrated a strong neuroprotective effect against the oxidative stress induced in the model (Figure 5A,B). The protective effects likely arise from both the preservation of the GSH levels and the scavenging of free radicals generated by *t*-BuOOH metabolism.

The compounds investigated in this evaluation were determined to preserve the GSH levels in the range of 60% to 80% compared to *t*-BuOOH alone. Regarding MDA production, the compounds reduced the levels by 29% to 41%, with **11** reducing the MDA production by 41%, all compared to *t*-BuOOH alone.

#### 2.5.3. Effect of Pyrrole Hydrazones in a Model of Non-Enzyme Lipid Peroxidation in Isolated Rat Brain Microsomes

The incubation of brain microsomes with Fe^2+^/AA (20 μM ferrous sulfate solution and 0.5 mM ascorbic acid solution) led to a substantial increase in MDA production by 268% compared to the untreated controls. However, when the test substances were co-incubated with Fe^2+^/AA at a concentration of 100 µM, they exhibited significant antioxidant effects, effectively mitigating the oxidative impact induced by Fe^2+^/AA (Figure 6).

In this non-enzymatic lipid peroxidation model, compound **1** decreased MDA production by 35%, compound **7** by 38%, compound **9** by 26%, compound **11** by 35%, compound **12** by 40%, compound **14** by 37%, and compound **15** by 36% compared to the Fe^2+^/AA group.

### 2.6. In Vitro Evaluation of the Neuroprotective Effects of the Target Pyrrole Hydrazones on Neuroblastoma SH-SY5Y Cells

SH-SY5Y cells were pre-treated with test compounds at concentrations of 0,1, 1, 5, 10, 20, and 50 μM, followed by exposure to H_2_O_2_ (1 mM for 10 min) (Figure 7). 

We found that all compounds showed a statistically significant neuroprotective effect in concentrations ranging from 0.1 to 50 µM. The best neuroprotection was observed at concentrations of 10 and 5 µM, where compounds **9**, **12**, and **14** showed the most pronounced effects (*p* < 0.05). At a concentration of 10 μM, these compounds exerted protective effects of 52%, 53%, and 51%. At a concentration of 5 μM, the protective effects were 47%, 48%, and 33%, respectively. Interestingly, compound **12** demonstrated strong protective effects even at the highest concentrations of 20 μM and 50 μM (by 50% and 32%, respectively).

## 3. Discussion

The selection of the investigated pyrrole-based hydrazones was done after taking into consideration the promising in vitro radical scavenging activities of a previously studied group of substances [11]. The effects were determined by spectrophotometric measurement of the discoloration of the stable free radical DPPH^•^ (2,2-diphenyl-1-picrylhydrazyl) at λ_max_ = 517 nm and, additionally, by evaluation of the reduction of the dark blue ABTS^•+^ radical cation to colorless ABTS (2,2-azino-bis (3-etilbenzotiazolin)-6-sulfonic acid) [22]. The most active derivatives to be further characterized were chosen based on the outcomes of the aforementioned study. Moreover, in the search of multi-targeted derivatives, compounds with MAOB and AChE inhibitory activities were also selected [21].

The necessary amounts of the target substances were resynthesized using the ecologically friendly MW-assisted approach. The structure of the obtained compounds was defined through an appropriate comparison-based TLC method, where the obtained R_f_ values defined the similarity.

Previous pharmacological studies of the selected derivatives showed that the introduction of bulky substituents in the azomethine fragment decrease the MAOB inhibitory activity and underline the AChE inhibitory effect. On the other hand, as expected, the presence of strong electron-withdrawing residues is a prerequisite for good radical scavenging effects.

The observed discrepancies in the performed biological effects of the tested molecules pointed our attention towards investigation of the possibility of the appearance of a relationship between the physicochemical properties of the derivatives and the neurotoxicity. Following this idea, we performed some preliminary screening of the tested derivatives on a variety of in vitro models, aiming to decrease the unnecessary testing. The results obtained indicate that there is no significant influence of the physicochemical properties of the azomethine residues on the effects of the molecules investigated in the models used. This gave us the opportunity to enrich the knowledge on the behavioral characteristics of the evaluated molecules to further identify a lead compound to be subjected to further testing.

The majority of marketed drugs reach their targets via passive diffusion across biological membranes. Gastrointestinal (GIT) absorption and the delivery of molecules to the CNS is highly dependent upon their molecular weight, lipophilicity, polar surface area, hydrogen bonding, and extend of ionization [23]. In the present work, prior to the experimental studies, the physicochemical properties of compounds related to GIT and BBB permeability were predicted in silico.

The results from the performed in silico calculations identified that the molecular weights range between 509.40 and 697.62 g/mol, which is a violation of the Lipinski’s rule and should be addressed further in the development process [24]. Compounds **1**, **7, 9**, **11**, **12**, and **14** fulfil the rest requirements of the classical rule, namely logP < 5, number of H-bond donor atoms < 5, and H-bond acceptor atoms < 10, which is prospective with respect to GIT permeability. All compounds, except **1** (amphoteric), are weak acids with pKa values between 8.38 and 11.17 and are unionized at pH 7.4. This is additionally confirmed by the match of logP and logD7.4 values. Low ionization in the physiological conditions and optimal lipophilicity are the prerequisites for good permeability. The logP values around 3 for most compounds also indicate their potential to become leads after proper optimization. TPSA values fall below the threshold of 140 Å^2^ (with the only exception of **11**), which is beneficial for their absorption, permeability, and oral bioavailability [25]. This is in line with previous studies with a pyrrole-based hydrazide, which was predicted to have high oral absorption and ability to cross the blood–brain barrier [26].

The anatomical and functional features of the BBB present higher and more stringent criteria to compounds regarding permeability [27,28]. With logP and logD7.4 between 3.02 and 5.17, lack of ionization, 2-3 H-bond donors (HBD), and up to 7 H-bond acceptors (HBAs), the compounds meet the criteria for optimal penetration. The TPSA values are slightly beyond the accepted limit (**11** is an exception). MW and flexibility of the molecules should be regarded as a topic of future optimization. Despite the number of free rotatable bonds exceeding the requirement, *p*-π conjugation in the fragment NH=N-C-planar aromatic structure makes that part of the molecule quite rigid.

The in silico screening differentiated compounds **1**, **7**, **9**, **12**, and **14** as candidates for further development. To additionally refine this selection, the neurotoxicity and neuroprotection of the target molecules were evaluated in vitro on subcellular and cellular models.

Neurotoxicity refers to damage to the brain and nervous system, caused by external and internal factors. It can result in structural and functional changes [29].

The high oxygen consumption of the brain, the abundance of polyunsaturated fatty acids, and the relative deficiency of antioxidant enzymes in the central nervous system together contribute to its increased vulnerability to lipid peroxidation [30]. Very often, the increased amounts of ROS are the result of cells’ metabolic activity [31]. This may result in a condition, known as oxidative stress, related to a reduced antioxidant capacity and an increased formation of ROS and RNS. The ensuing imbalance results in the modification of cellular components, especially lipids [32]. The extent of lipid peroxidation plays an important role in pathology conditions associated with oxidative stress. The important role of free radical-induced lipid peroxidation has been reported to be a major factor for a variety of pathological processes.

Thus, due to the complex methodology required for direct in vivo quantitation of free radicals, the application of a suitable model for indirect measurement of the effects of the free radicals on the cellular components that they modify, such as proteins [32] and lipids [33], is preferred.

We evaluated the in vitro toxicity of the newly synthesized compounds using various subcellular and cellular models. Cell-based systems and subcellular fractions have long been used as in vitro models in neurotoxicological research. These systems are suitable for studying signaling pathways and receptor-mediated signal transduction, allowing specific mechanisms to be examined outside the cellular environment [34]. We found that the test compounds caused mild toxic effects on the different subcellular fractions used in the study.

Synaptosomes are subcellular fractions derived from neurons and prepared from brain tissue through homogenization. They function as small, anucleated entities that retain neuronal vesicles, enzymes, mitochondria, and highly active ion transport systems across their membranes [34]. The toxicity evaluation of the selected pyrrole-containing compounds was studied on isolated rat brain synaptosomes, and the results showed that compound **12** may be considered as the least toxic. This observation highlights a similarity to a study of another, structurally related series of hydrazones, in which the presence of a 2-hydroxyphenyl substituent is associated with lower synaptosomal toxicity [20].

The isolated brain mitochondria and microsomes are a crucial in vitro tool in neurotoxicology. They retain many functional properties similar to those in natural in situ and in vivo environments [34]. Malondialdehyde (MDA) is one of the best known secondary products of lipid peroxidation, and it can be used as a marker of a cell membrane [33], since its amount has often been pointed to as a main parameter for the evaluation of lipid peroxidation and/or injury mediated by free radicals release [32]. Thus, MDA is by far the most popular indicator of oxidative damage to cells and tissues. In this study, we observed a decreased MDA production in oxidative damage models with isolated brain microsomes, treated with the target pyrrole–hydrazones. In the absence of a prooxidant agent, most compounds cause a weak increase in MDA production, which is, however, outweighed by their stronger antioxidant effects. Similar hormetic behavior has been observed with well-known nutritional antioxidants [35]. Thus, our results might be interpreted as an indication of protective radical scavenging capacity. The antioxidant effect in vitro, observed with the test substances, is likely due to their ability to scavenge free radicals.

The toxicological characterization was also performed at the cellular level. Neuroblastoma cell line SH-SY5Y is an appropriate in vitro model system in neurotoxicity studies, especially for dopaminergic neurons [36]. Neuronal characteristics and marked sensitivity to oxidative stress make it a suitable model for studying neurological pathologies at the molecular, morphological, and physiological levels [37]. All of the compounds, **7**, **9**, **12**, **14**, and **15**, exerted protective effects, which were comparable in magnitude to observed effects in another published study with structurally related compounds [20].

A characteristic feature of this cell line is the pronounced susceptibility to oxidative stress, the cause of which may be the increased formation of reactive radical species associated with dopamine synthesis. Our results showed that the introduction of bulky aromatic residues (compounds **14** and **15**) is associated with a decrease in the cytotoxicity, while the appearance of active functional groups such as –NO_2_ (in compound **11**) and –COOH (in compound **1**) are related to a slight increase in the cytotoxicity.

Next, the protective effects of the target compounds were extensively evaluated in different in vitro models, among which are 6-OHDA-induced neurotoxicity, which mimics the neurodegenerative processes in Parkinson’s disease, and *t*-BuOOH, which induces oxidative stress. The latter is characterized by a depletion of reduced glutathione (GSH) and oxidation of -SH groups in key mitochondrial enzymes, accompanied by disruption of the mitochondrial membrane integrity due to the action of reactive oxygen species (ROS), which ultimately leads to lipid peroxidation [38,39,40]. The results from those studies identified strong neuroprotective effects of the selected compounds on SH-SY5Y cells at the applied concentration interval. Noteworthy, compound **12** showed high protective activity at concentrations 5, 10, 20, and 50 μM. It is worth mentioning that, at the highest applied treatment concentration, compound **12** showed the lowest observed protection (32%), while, at the other three concentrations, the protective effect was comparable (around 50%).

Considering the basis of the performed experiments and the fact that the used parameter evaluation is pointed mainly towards the ability of the compounds to interact or scavenge free electrons, it has come to our attention that the most promising results are obtained by evaluating compounds **7** and **12**, containing a phenolic hydroxyl group with well-determined radical scavenging properties. On the other hand, the presence of the highly aromatic naphthalene core in molecule **14** is a prerequisite for improved electron transfer properties and electron capture abilities, which correlates with the promising observed neuroprotective results. The slight decrease in the protective effects of compound **7** compared to **12** is probably due to the appearance in the molecule of the bulky methoxy groups in the m-position of the final phenyl fragment.

It is worth noting that the introduction of the two synaptosomes were centrifuged –OCH_3_ groups in the structure of compound **7** causing a slight decrease in the observed neuroprotective effects and for compound **7** to be less protective than its analog, **12**. This may be due to structural hindrance of the availability of the free –OH group, strongly related to the expected appearance of antioxidant properties.

The neuroprotective effects of the newly synthesized Schiff base compounds were also investigated in vitro by a H_2_O_2_-induced oxidative stress model in neuroblastoma SH-SY5Y cells. Direct incubation of cells with H_2_O_2_ is an in vitro method widely used to induce oxidative stress. Hydrogen peroxide is a reactive oxygen species (ROS) and is involved in the generation of other ROS via the Fenton and Haber–Weiss reactions [41]. Hydrogen peroxide can also cross biological membranes [42]. In the model of H_2_O_2_-induced oxidative stress, all compounds exhibited significant neuroprotection, with the strongest effects observed at 5 and 10 μM. Compounds **9**, **12**, and **14** were the most potent, providing 52%, 53%, and 51% protection at 10 μM and 47%, 48%, and 33% at 5 μM, respectively. Notably, compound **12** retained its effectiveness at 20 μM (50%) and 50 μM (32%).

The results from the performed neuroprotective evaluations are consistent with the results obtained previously on the discussed in 19 in vitro radical scavenging effects of the selected hydrazones. The data indicated that the presence of a hydroxyl group is a prerequisite for the manifestation of good radical scavenging properties. On the other hand, the introduction of bulky substituents or the blocking of the free hydroxyl group reduces both the antioxidant properties of the studied molecules and their neuroprotective effects, determined by the parameters we investigated.

The in silico screening prioritized compounds **1**, **7**, **9**, **12**, and **14** as the best candidates for further development. To additionally refine this selection, both the physicochemical characteristics of the compounds and the experimental results were analyzed. Data from the in vitro studies were normalized by a scaling procedure that assigns activity to the range 0–1 and makes a clear distinction between active and inactive compounds Appendix A. Since compounds **1** and **11** showed moderate cytotoxicity, they were discarded from further analyses. Compounds **7**, **12**, and **14** showed better results in the subcellular models for both cytotoxicity and neuroprotection. When tested in the SH-SY5Y neuroblastoma cell line, compounds **9**, **12**, and **14** exerted the best neuroprotection. The poorer neuroprotection of compound **7** may be attributed to its limited diffusion across the cell membrane. In fact, compound **7** has a higher MW and number of HBA atoms than compounds **9**, **12**, and **14**. Its molecule is the most flexible (RB = 14) and is the least lipophilic compound in the selection.

These findings are promising, because they highlight the potential applications of the new compounds in mitigating oxidative stress-related neuronal damage. More in-depth studies are needed to elucidate their role in neuroprotection.

## 4. Materials and Methods

### 4.1. Chemistry

The applied solvents and reactants were obtained from commercial suppliers (Sigma-Aldrich, Darmstadt, Germany and Fluka Buchs, Switzerland ) and used without further purifications. The microwave-assisted synthetic reactions were performed in a FlexiWave Milestone Lab Microwave reactor. The completion of the chemical reactions was monitored by TLC. The infrared spectra (4000–400 cm^–1^ range) of the resynthesized compounds were recorded on a Nicolet iS10 FTIR spectrometer, Smart iTR adapter (OMNIC version 8.0 Thermo Fisher Scientific, Waltham, Massachusetts, MA 02115 USA). Thereafter, the spectra were compared with the reported data.

### 4.2. In Silico Calculation of Physicochemical Properties

Most of physicochemical properties of the compounds were calculated by using the SwissADME tool [43,44].

In addition, the pKa DB tool of ACD Labs software v. 9.08 (ACD/Labs, Toronto, ON, Canada) was used for the pKa calculations.

The theoretical Log D 7.4 was calculated based on the following expressions:

LogD7.4 = LogP − log (1 + 10^(pH−pKa)^) for acids and LogD7.4 = LogP − log (1 + 10^(pKa−pH)^) for bases.

### 4.3. Experimental Animals

The experiments involved 2 animals obtained from the National Breeding Centre of the Bulgarian Academy of Sciences. The animals were housed under standard conditions in plexiglass cages with free access to water and food. Food was withdrawn 12 h before each specific study. The experiments were conducted in accordance with Ordinance No. 15 on Minimum Requirements for the Protection and Welfare of Experimental Animals (SG No. 17, 2006) and the European Regulation for the Handling of Experimental Animals. The study was approved by the Bulgarian Food Safety Agency, with permission No. 273, valid until 20 July 2025.

### 4.4. Preparation of Rat Brain Synaptosomes and Mitochondria

Synaptosomes and mitochondria were isolated by subcellular fractionation using a Percoll gradient, following the method described by Tauplin et al. [45] and Sims et al. [46]. Brain homogenate was prepared and centrifuged at 1000× *g* for 5 min at 4 °C. The supernatants were collected and centrifuged again at 1000× *g* for 5 min at 4 °C. The supernatants from both centrifugations were combined and distributed into four tubes. These tubes were then centrifuged at 10,000× *g* for 20 min at 4 °C three times. The final two centrifugations were performed to purify the synaptosomes and mitochondria.

### 4.5. Isolation of Synaptosomes and Mitochondria

The isolation process involves using a colloidal silicon solution, specifically Percoll. First, a 90% stock solution of Percoll is prepared. Next, Percoll solutions at concentrations of 16% and 10% are prepared, with 4 mL of each solution placed into six separate test tubes. To the precipitate from the final centrifugation step, 7.5% Percoll (derived from the 90% stock solution) is added. The tubes are then centrifuged at 15,000× *g* for 20 min at 4 °C. This procedure effectively isolates synaptosomes and mitochondria. After centrifugation, three distinct layers form in the tubes. The bottom layer contains mitochondria, the top layer contains lipids, and the middle layer—located at the 16% to 10% Percoll interface—contains synaptosomes. The corresponding layer from each tube is carefully removed using a glass Pasteur pipette and combined into one. Buffer B with glucose is then added to the mixture, which is centrifuged at 10,000 × g for 20 min at 4 °C, allowing the isolation buffer to be exchanged with the incubation buffer. After centrifugation, the pellet containing the synaptosomes is resuspended in Buffer B with glucose. Synaptosomes and mitochondria were incubated with the test substances at a concentration of 100 µM for 1 h [39,46].

### 4.6. 6-OHDA-Induced Neurotoxicity

This in vitro model mimics the neurodegenerative processes primarily observed in Parkinson’s disease (PD). The metabolism of 6-OHDA produces reactive quinones (p-quinone), which, in turn, generate reactive oxygen species (ROS). These reactive metabolites and ROS cause damage to both pre- and post-synaptic membranes, ultimately leading to neuronal cell damage [47]. In this study, synaptosomes were incubated with 6-OHDA (150 μM) for 1 h to establish the model [48].

### 4.7. MTT Assay to Assess the Synaptosomal Viability

After a 1-h incubation with the substances and toxic agent, synaptosomes were centrifuged in a microcentrifuge at 15,000× *g* for 1 min. The pellet was gently mixed with Buffer B + glucose, and the supernatant containing 6-OHDA was discarded to prevent oxidation of 3-(4,5-dimethylthiazol-2-yl)-2,5-diphenyltetrazolium bromide, a tetrazole (MTT). The synaptosomes were centrifuged again at 15,000× *g* for 1 min. After the second wash, Buffer B + glucose was added to the pellet. Then, 60 µL of MTT solution was added to the “washed” synaptosomes. The plates were incubated with the MTT solution at 37 °C for 10 min. After incubation, the samples were centrifuged at 15,000× *g* for 2 min. The excess liquid was removed, and a DMSO solution was used to dissolve the formed formazan crystals. The amount of formazan was then measured spectrophotometrically at λ = 580 nm [49].

### 4.8. Determination of Reduced Glutathione (GSH) in Isolated Brain Synaptosomes

Following protein precipitation with trichloroacetic acid, the thiol groups in the supernatant were measured using DTNB, which produces a yellow-colored compound that absorbs light at λ = 412 nm. After incubation, synaptosomes were centrifuged at 4000× *g* for 3 min. The supernatant was removed, and the pellet was collected for GSH determination. The pellet was treated with 5% trichloroacetic acid and left on ice for 10 min. It was then centrifuged at 8000× *g* for 10 min at 2 °C. The supernatant was collected for GSH determination and frozen at −20 °C. Immediately before measurement, the samples were neutralized with 5N NaOH [50].

### 4.9. Tert-Butyl Hydroperoxide-Induced Oxidative Stress

Isolated brain mitochondria were incubated with 75 µM *tert*-butyl hydroperoxide (*t*-BuOOH) [51].

### 4.10. Determination of Malondialdehyde (MDA) Production in Brain Mitochondria

To the mitochondria, 0.3 mL of 0.2% thiobarbituric acid and 0.25 mL of 0.05 M sulfuric acid were added, and the mixture was boiled for 30 min. After boiling, the tubes were placed on ice, and 0.4 mL of n-butanol was added to each. The samples were then centrifuged at 3500× *g* for 10 min. The amount of MDA was determined spectrophotometrically at 532 nm [52].

### 4.11. Determination of the GSH Level in Brain Mitochondria

After incubating the mitochondria with the substances and tert-butyl hydroperoxide, the reaction was stopped by adding 5% trichloroacetic acid. Each sample was then homogenized with the acid and left on ice. Following centrifugation of the homogenate at 6000× *g*, a 0.04% solution of DTNB was added to the supernatant, producing a yellow color. The absorbance was measured spectrophotometrically at 412 nm [52].

### 4.12. Isolation of Brain Microsomes

The brain was homogenized in 9 volumes of 0.1 M Tris buffer containing 0.1 mM dithiothreitol, 0.1 mM phenylmethylsulfonyl fluoride, 0.2 mM EDTA, 1.15% KCl, and 20% (*v*/*v*) glycerol (pH 7.4). The resulting homogenate was centrifuged twice at 17,000× *g* for 30 min. The supernatants from both centrifugation steps were combined and centrifuged twice at 100,000× *g* for 1 h. The pellet was then frozen in 0.1 M Tris buffer [53].

### 4.13. Iron-/Ascorbate-Induced Lipid Peroxidation (LPO)

Non-enzyme-induced lipid peroxidation was induced with 20 μM ferrous sulfate solution and 0.5 mM ascorbic acid solution [54].

### 4.14. Determination of MDA in Brain Microsomes

After incubating the microsomes with the substances and toxic agent, the reaction was stopped by adding 0.5 mL of 20% trichloroacetic acid, followed by 0.5 mL of 0.67% thiobarbituric acid. The reaction formed a colored complex between the malondialdehyde produced and thiobarbituric acid. The amount of malondialdehyde (MDA) was measured spectrophotometrically at 535 nm, using a molar extinction coefficient of 1.56 × 10^5^ M^−1^ cm^−1^ for the calculation [54].

### 4.15. Cell Cultures and Treatment

The neuroblastoma cell line SH-SY5Y was acquired from Sigma-Aldrich (ACACC cell lines). They were cultured in Roswell Park Memorial Institute 1640 (RPMI) cell medium supplemented with 10% heat-inactivated fetal bovine serum (FBS) and 2 mM L-glutamine. The cells were maintained at 37 °C in an atmosphere containing 5% CO_2_ in a humidified incubator.

SH-SY5Y cells were seeded in 96-well plates at a density of 2.5 × 10^4^ cells/well. The plates were incubated for 24 h to ensure proper attachment. Cell viability assays were performed after treatment with the test compounds at concentrations of 0.1, 1, 5, 10, 50, 100, 250, and 500 μM, according to the protocol of Mosmann (1984) [55]. Each compound was tested in 6 wells. Following the treatment for 24 h, a solution of (3-(4,5-dimethylthiazol-2-yl)-2,5-diphenyltetrazolium bromide) MTT at a concentration of 5 mg/mL was prepared and briefly vortexed. The culture medium was aspirated and replaced with 100 μL of MTT solution in each well. The plates were then incubated for 3 h. After incubation, the formazan crystals were dissolved using 100 μL dimethylsulphoxide (DMSO), and the absorbance was measured at 570 nm (690 nm for the background absorbance) using a multiple reader Synergy 2 (BioTek Instruments, Inc., Highland Park, Winooski, VT, USA).

In the oxidative stress model, SH-SY5Y cells (3.5 × 10^4^ cells/well) were exposed to 1 mM H_2_O_2_ for 10 min. The cells were pre-treated for 1.5 h with the test compounds (0.1, 1, 5, 10, 20, and 50 µM). Subsequently, a H_2_O_2_ solution was added. Cell viability was assessed after 24 h by the MTT-dye reduction assay, as described above.

### 4.16. Statistical Analysis

The results of the experiments performed on isolated brain synaptosomes, mitochondria, and microsomes were statistically processed using the ‘MEDCALC’ program using the non-parametric Mann–Whitney method at significance levels of *p* < 0.05, *p* < 0.01, and *p* < 0.001. Statistical analyses with the SH-SY5Y cell line were conducted on GraphPad Prism software (version 8, GraphPad Software, La Jolla, CA, USA). The significance of the data was assessed using one-way analysis of variance (ANOVA), followed by Dunnet’s test for post hoc multiple comparisons, to evaluate statistical differences. Values of * *p* ≤ 0.05, ** *p* ≤ 0.01, and *** *p* ≤ 0.001 were considered statistically significant. Cytoprotecting data were normalized as a percentage of the H_2_O_2_-treated control (defined as 0% viability). Statistical analysis was performed to compare the different groups of compounds. Statistical significance was determined using the Holm–Sidak method, with alpha = 0.05. Each row was analyzed individually, without assuming a consistent SD.

## 5. Conclusions

The current extensive in vitro study of the resynthesized pyrrole-based compounds with proven radical scavenging, MAO-B, and AChE inhibitory effects revealed that all test compounds displayed low neurotoxicity with strong antioxidant and neuroprotective effects. In the SH-SY5Y cell model, compounds **7**, **9**, **12**, **14**, and **15** exhibited low toxicity and significant neuroprotection. In silico screening further identified compounds **7**, **9**, **12**, and **14** as candidates for further development. Thus, the tested pyrrole-based compounds could serve as a promising basis for the development of potential neurodegenerative disease therapeutics, warranting further research on understanding their mechanisms of action and optimizing their efficacy and therapeutic potential.

## Data Availability

Data is contained within the article.

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
