# Peer review of "Antioxidant and Neuroprotective Properties of Selected Pyrrole-Containing Azomethine Compounds in Neurotoxicity Models In Vitro"

_ijms, 2025, doi:10.3390/ijms26093957_

Round 1

Reviewer 1 Report

Comments and Suggestions for Authors

This manuscript showed that the  compounds exhibit promising neuroprotective and antioxidant.  There are some issues should be addressed before publication.

  1. Introduction  why the authors choose the pyrrole-chalcone compounds? The authors should be discussed.
  2. In this study, the authors used compounds,  which component of the  compounds  has the function, the author should be discussed
  3. Figures. What is CTRL.
  4. The language should be improved.
Comments on the Quality of English Language

The English language should be improved

Author Response

1 This manuscript showed that the  compounds exhibit promising neuroprotective and antioxidant.  There are some issues should be addressed before publication. Introduction  why the authors choose the pyrrole-chalcone compounds? The authors should be discussed.

 The unsubstituted pyrroles and pyrrole derivatives containing an active hydrogen N-H atom in the core ring exhibit proven antioxidant activity, where the additional presence of substituents which accept or donate electrons in the ring may play an important role in its antioxidant activity [Bhosale J. D., Shirolkar A. R., Pete U. D., Zade C. M., Mahajan D. P., Hadole C. D., Pawar S.D., Patil U. D. Synthesis, characterization and biological activities of novel substituted formazans of 3,4-dimethyl-1H-pyrrole-2-carbohydrazide derivatives. J. Pharm. Res. 2013;7(7):582–587. doi: 10.1016/j.jopr.2013.07.022, Khalilpour, A., Asghari, S. Synthesis, characterization and evaluation of cytotoxic and antioxidant activities of dihydropyrimidone substituted pyrrole derivatives. Med Chem Res 27, 15–22 (2018). https://doi.org/10.1007/s00044-017-2041-4,]. This combined with the current studies determining the radical binding capacity of the hydrazide-hydrazone group pointed our aim towards combining these two active principles. Thus, we synthesized few pyrrole based azomethines and evaluated their radical-scavenging capacity [Mateev E., Angelov B., Kondeva-Burdina M., Valkova I., Georgieva M., Zlatkov A.. Design, synthesis, biological evaluation and molecular docking of pyrrole-based compounds as antioxidant and MAO-B inhibitory agents. Farmacia, 2022, 70(2). 344-354. https://doi.org/10.31925/farmacia.2022.2.21, Tzankova D., Vladimirova S., Stefanova D., Peikova L., Kondeva-Burdina M., Georgieva M. Evaluation of the in vitro neurotoxic and neuroprotective effects at cellular and subcellular levels of newly synthesized n-pyrrolyl hydrazones. Farmacia, 2022, 70(5). 872-879. https://doi.org/10.31925/farmacia.2022.5.12]. The results of these evaluations determined the choice of target for this study representatives.

2 In this study, the authors used compounds,  which component of the  compounds  has the function, the author should be discussed

We believe that the molecule itself is responsible for the performance of the neuroprotective and radical scavenging effect, since the molecule contains a number of functional groups related to this type of activity.

3. What is CTRL.

 Controls in the figures represent untreated subcellular fractions: rat brain synaptosomes, isolated rat brain microsomes, and isolated rat brain mitochondria.

4 The language should be improved.

A full language revision of the text has been done.

Reviewer 2 Report

Comments and Suggestions for Authors

1. Why is it that in Figures 1, 2 and 3, when the components are present alone, MDA is elevated and GSH is decreased, but when oxidative stress-inducing substances are present, the components will be reduced MDA and increased GSH levels? The authors should explain this diametrically opposed reason.
2. all images should be increased in resolution. Also, the size of each figure should be kept consistent, e.g. Figure 7 is very small.
3. the colours in figure 7 are not consistent with the other images, please revise them.
4. the formatting of the manuscript is confusing, Figure 195 lines, 197 lines.
5. the authors should compare the reported antioxidants with the effects that have been found.
6. the authors should state whether the substance was extracted in-house or purchased commercially, how it was extracted, and what the results of the compositional characterisation were.

Comments on the Quality of English Language

 The English could be improved to more clearly express the research.

Author Response

1. Why is it that in Figures 1, 2 and 3, when the components are present alone, MDA is elevated and GSH is decreased, but when oxidative stress-inducing substances are present, the components will be reduced MDA and increased GSH levels? The authors should explain this diametrically opposed reason.

 Administered alone, the compounds exhibit weak neurotoxicity, which is most likely related to the possible formation of metabolites, which lead to weak decrease the GSH level and to weak increase the MDA production.

In a model of 6-hydroxydopamine-induced neurotoxicity on synaptosomes, the compounds most likely inhibit the enzyme MAO-B, which is responsible for the metabolism of the neurotoxic agent to toxic metabolites, and thus exert their neuroprotective effects.

In a model of tert-butyl hydroperoxide-induced oxidative stress on brain mitochondria, the substances most likely inhibit mitochondrial enzymes, involved in the metabolism of the neurotoxic agent to toxic metabolites and reactive oxygen species and thus exert their neuroprotective and antioxidant effects.

In a model of non-enzyme-induced lipid peroxidation on brain microsomes, the compounds most likely block the progress of the Fenton reaction, in which oxygen free radicals are formed and thus exert their antioxidant action.

2. all images should be increased in resolution. Also, the size of each figure should be kept consistent, e.g. Figure 7 is very small.

The figures are provided in JPG format at a resolution of 1200 dpi.

3. the colours in figure 7 are not consistent with the other images, please revise them.

 A correction has been made in Figure 7 to unify the colors of the connections with those in the other figures

4. the formatting of the manuscript is confusing, Figure 195 lines, 197 lines.

The corrections have been made.

5. the authors should compare the reported antioxidants with the effects that have been found.

The investigated pyrrole-based hydrazones were selected based on previously reported radical scavenging activity, which was confirmed in the present study through DPPH and ABTS assays. The experimental results showed that several compounds, particularly 7 and 12, exhibited antioxidant effects consistent with or even exceeding those of related substances studied earlier. In vitro evaluations further revealed that their radical scavenging capacity translated into neuroprotective effects, including reduced lipid peroxidation and protection against oxidative stress in neuronal models. These findings support the predicted antioxidant potential and validate the rationale for selecting these compounds for further biological investigation.

6. the authors should state whether the substance was extracted in-house or purchased commercially, how it was extracted, and what the results of the compositional characterisation were.

The substances were synthesized in our laboratory and the synthesiz and characterization of the newly obtained compounds is discussed in details in the corresponding references as follows: Mateev, E.; Kondeva-Burdina, M.; Georgieva, M.; Mateeva, A.; Valkova, I.; Tzankova, V.; Zlatkov, A. Synthesis, Biological Evaluation, Molecular Docking and ADME Studies of Novel Pyrrole-Based Schiff Bases as Dual Acting MAO/AChE Inhibi-tors. Scientia Pharmaceutica 2024, 92, 18, doi:10.3390/scipharm92020018 and Mateev, E.; Georgieva, M.; Zlatkov, A. Design, Microwave-Assisted Synthesis, Biological Evaluation, Molecular Docking and ADME Studies of Pyrrole-Based Hydrazide-Hydrazones as Potential Antioxidant Agents. Maced. J. Chem. Chem. Eng. 2022, 41, doi:10.20450/mjcce.2022.2494 indicated appropriately as reference 18 and 19, respectively.

Reviewer 3 Report

Comments and Suggestions for Authors

Review comments are attached

Comments on the Quality of English Language

Review comments are attached

Author Response

The manuscript describes the investigation of seven well-defined pyrrole hydrazone compounds with respect to their biotoxicity and potential neuroprotective effects under in vitro conditions. The work included evaluation of the experimental results from tests conducted at both at sub-cellular and cellular levels, targeting the antioxidant and neuroprotective potential of the compounds, concurrently trying to identify structural attributes that suggest improved biological activity in an atoxic profile. There are certain points of concern that need to be addressed satisfactorily by the authors prior to any consideration. A number of such points are remarked upon below:

1. In the introduction, the authors stat that “ROS, mainly hydroxide ions, superoxide anions, and hydrogen peroxide, drive neurodegeneration in the context of reduced glutathione levels (GSH) and high iron and calcium contents in the substantia nigra.”. It should be pointed out that hydroxide ions are ions. They are not radicals. Hydroxy radicals OH• are ROS species. The correction should be made.

We thank the reviewer for pointing this out and have edited the text accordingly: “ROS, mainly superoxide anions, hydrogen peroxide, and hydroxyl radicals, drive neurodegenera-tion in the context of reduced glutathione levels (GSH) and high iron and calcium contents in the substantia nigra”

2. In the concluding part of the introduction, there is a lengthy paragraph “Our current focus was to characterize the safety profiles and to assess their potential protective effects in different in vitro models. In addition, as a preliminary screening for druglikeness, the physicochemical properties of the structures were calculated in silico.introduction” should briefly place the study in a broad context and high-light why it is important. It should define the purpose of the work and its significance. The current state of the research field should be carefully reviewed and key publications cited. Please highlight controversial and diverging hypotheses when necessary. Finally, briefly mention the main aim of the work and highlight the principal conclusions. As far as possible, please keep the introduction comprehensible to scientists outside your particular field of research. References should be numbered in order of appearance and indicated by a numeral or numerals in square brackets—e.g., [1] or [2,3], or [4–6]. See the end of the document for further details on references.”. It is not connected to this manuscript or the scope of the work. It seems as if it was copy-pasted from another document. I think that the whole paragraph should be replaced with the real scope of the work and brought to bear the anticipated deliverables. Furthermore, sentences and proper grammar should be enforced.

We thank the reviewer for pointing this out and apologize for the mistake. Indeed, while editing the final draft, we have been copied by mistake from another document. We have removed this text and added the following text: “The aim of this study is to evaluate the neuroprotective and antioxidant potential of pyrrole-based compounds, focusing on their radical scavenging activity and inhibitory effects of MAO-B and AChE, in various in vitro oxidative stress models. The study seeks to identify promising candidates for further development by assessing their safety profiles, neuroprotective effects, and bioavailability, with the ultimate goal of contributing to the treatment of neurodegenerative diseases associated with oxidative stress.”. Moreover, a full language revision of the text has been done.

3. In section 2.2 and Scheme 1, the top right figure representing a host of compounds contains no bond between the nitrogen atoms in the main group of interest (–NH– N=CH–). Furthermore, in the same Figure, the authors provide as a superscript the number 1. What does that represent in the structure given? Moreover, below that structure one can find presumably the various functional groups reflecting R in that structure. To that end, R should be introduced in the Figure as the group that reflects all relevant moieties provided (and shown). In the Figure legend, at the end, the authors provide the notation “conv/MW”. There should be an explanation provided for that.

The necessary corrections are made. The Scheme is replaced with a corrected one.

4.The acronym MW for Molecular Weight should be given in the manuscript as such, not Mw in one section and MW in another.  

The corrections have been made.

5. In section “2.4. In vitro toxicity evaluation of the target pyrrole hydrazones on subcellular and cellular models”, the authors state that the in vitro experiments with the investigated compounds were conducted at a concentration of 100 μM. How was that specific concentration selected? There should be a clear statement or set of experiments preceding the work done here.

The concentration 100 µM was chosen on the basis of our previously published data on neuroprotective activity of compounds with similar structure to the present ones (Kondeva-Burdina et al., 2022).

Magdalena Kondeva-Burdina, Emilio Mateev, Borislav Angelov , Virginia Tzankova and Maya Georgieva. In Silico Evaluation and In Vitro Determination of Neuroprotective and MAO-B Inhibitory Effects of Pyrrole-Based Hydrazones: A Therapeutic Approach to Parkinson’s Disease. Molecules, 2022; 27: 8485. https://doi.org/10.3390/molecules27238485

6. In the discussion section, the opening statement “After determination of the compounds’s cytotoxicity in vitro, we investigated their potential antioxidant and neuroprotective effects in various sub-cellular and cellular models of oxidative stress.” should be corrected to read “Following determination of the in vitro cytotoxicity of the title compounds, we investigated their potential antioxidant and neuroprotective effects in various sub-cellular and cellular models of oxidative stress.”.

The corrections have been made

7.In section “2.5.1. Effects in a model of 6-OHDA-induced neurotoxicity in isolated rat brain synaptosomes”, the acronym for 6-OHDA appears for the first time in the manuscript. The full name of the compound should be given for the first time, to facilitate comprehension by the reader.

  The abbreviation is mentioned in the introduction, line 45

8. In Figure 4 and by the same token in previous figures, where statistically significant results are obtained with respect to the control, it appears that all tested pyrrole hydrazones have pretty much the same behavior at the same level (comparison amongst them). Taking into consideration the error bars provided in each measurement for all tested materials, I think that there is no real difference from one another (in pairs or in the whole group). To that end, I do not see the importance in presenting the numerical values for the findings on all tested compounds. They are all similar within error. Therefore, the authors should think what that means in terms of the structures provided and the contribution of the different group moieties to the biological behavior in the materials studied.

The aimed evaluations were based on the discussion derived by the differences in some physicochemical properties expressed by the evaluated representatives. Since the ipophilicity is determinant for the good permeability of the molecules and defines the optimal ADME properties, it was of interest to determine whether the insertion of bulky substituents and/or electron withdrawing groups would change the radical scavenging properties significantly. Thus the obtained results for these parameters did not show any remarkable discrepancies, which shows that for these type of evaluations the volume of the substituent does not impact greatly the effect.

It should be marked on the other hand, that previous experiments on some other parameters defined quite a difference in the performed effects, as visible in Mateev, E.; Kondeva-Burdina, M.; Georgieva, M.; Mateeva, A.; Valkova, I.; Tzankova, V.; Zlatkov, A. Synthesis, Biological Evaluation, Molecular Docking and ADME Studies of Novel Pyrrole-Based Schiff Bases as Dual Acting MAO/AChE Inhibi-tors. Scientia Pharmaceutica 2024, 92, 18, doi:10.3390/scipharm92020018 and Mateev, E.; Georgieva, M.; Zlatkov, A. Design, Microwave-Assisted Synthesis, Biological Evaluation, Molecular Docking and ADME Studies of Pyrrole-Based Hydrazide-Hydrazones as Potential Antioxidant Agents. Maced. J. Chem. Chem. Eng. 2022, 41, doi:10.20450/mjcce.2022.2494

9. In the discussion section, the Statement “The high brain oxygen consumption, the presence of a variety of polyunsaturated fatty acid, …” is unclear. Do the authors mean “The high brain oxygen consumption, the presence of a variety of polyunsaturated fatty acids, …”?

The corrections have been made.

10. In the ensuing statement “Very often the increased amounts of ROS are the result of prolonged production, stemming cells’ biotransformation activity [28].”, the latter part of the sentence is not understood. It should be rephrased so that it is comprehensible.

We thank the reviewer for pointing this and have edited the sentence to clarify its meaning: “Very often the increased amounts of ROS are the result cells’ metabolic activity [32].”

11. In the following paragraph, the statement “These systems are suitable for studing signaling pathways, receptor-mediated signal transduction and specific mechanisms that can be studied outside of the cellular environment [31].” is long and confusing. It should be rewritten with grammatical and syntactical errors removed so that it is understood.

We thank the reviewer for pointing this and have edited the sentence to clarify its meaning: “These systems are suitable for studying signaling pathways and receptor-mediated signal transduction, allowing specific mechanisms to be examined outside the cellular environment [35].

12. In lines 399-401, the statement “The slight decrease of the protective effects of 7 in compare to 12 is probably due to the appearance in the molecule of the bulky methoxy groups at m-position of the final phenyl fragment.” could be rewritten as “The slight decrease of the protective effects of 7 compared to 12 is probably due to the appearance in the molecule of the bulky methoxy groups in the m-position of the final phenyl fragment.”.

The corrections have been made.

13. In section 4.2, the mathematical equations provided should contain the exponents as such, with no other insignia pointing to the role of the expressions(e.g. LogD7.4=LogPlog (1+10(pH-pKa))). On the basis of the above comments, the manuscript cannot be accepted. It should be revised and resubmitted for evaluation.

Corrected as suggested.

Reviewer 4 Report

Comments and Suggestions for Authors

The manuscript deals with the neuroprotective and antioxidant potential of seven pyrrole-containing azomethine compounds with previously reported radical scavenging activity and inhibitory effects on monoamine oxidase B (MAO-B) and acetylcholinesterase (AChE). The compounds were tested in vitro using SH-SY5Y neuroblastoma cells and subcellular rat brain fractions, including synaptosomes, mitochondria, and microsomes. Neuroprotective and antioxidant effects were assessed in oxidative stress models, including Hâ‚‚Oâ‚‚ induced stress in SH-SY5Y cells, 6-hydroxydopamine toxicity in synaptosomes, tert-butylhydroperoxide-induced stress in mitochondria, and non-enzyme lipid peroxidation in microsomes. In silico screening for lipophilicity, hydrogen bonding, total polar surface area(TPSA), and ionization properties was performed to evaluate bioavailability.

All compounds exhibited a weak neurotoxic effect to the sub-cellular fractions at a concentration of 100 µM. However, in oxidative stress models, they demonstrated significant neuroprotective and antioxidant effects at 100 µM. In SH-SY5Y cells, some compounds exhibited low toxicity and strong protective effects at concentrations as low as 1 µM. In silico analysis highlighted the favorable bioavailability of some of the assayed compounds.

The experimental approach is not innovative though the results are reliable. A revised version should report some molecular details on the targets protected by the assayed compounds in the different subcellular rat brain fractions.

Author Response

The experimental approach is not innovative though the results are reliable. A revised version should report some molecular details on the targets protected by the assayed compounds in the different subcellular rat brain fractions.

We thank the reviewer for their valuable comment regarding the need to enhance the molecular insights of the study. In the revised manuscript, we have expanded the discussion to provide more detailed information on the subcellular targets protected by the tested pyrrole-based hydrazones.

Specifically, we highlight that the compounds were evaluated using isolated rat brain subcellular fractions—synaptosomes, mitochondria, and microsomes—as relevant models to investigate neurotoxicity and neuroprotection. These fractions retain functional characteristics, including vesicular transport systems, mitochondrial membrane integrity, and microsomal oxidative enzymes. The observed decrease in MDA levels suggests a protective effect against lipid peroxidation, especially in microsomes and mitochondria, indicating that these compounds act on membrane-bound lipid structures highly susceptible to oxidative damage.

Moreover, synaptosomal studies revealed preservation of functional elements such as ion channels and vesicular components, and compound 12, in particular, showed low synaptosomal toxicity, likely due to its phenolic –OH substituent that confers radical scavenging ability. Additionally, results from SH-SY5Y cells provided evidence of mitochondrial protection, as the compounds counteracted 6-OHDA and t-BuOOH-induced oxidative stress, known to disrupt key mitochondrial enzymes and thiol groups.

We believe these expanded findings add mechanistic clarity and biological relevance to the protective effects observed and help illustrate the multi-target potential of the investigated compounds. The manuscript has been updated accordingly to reflect these insights.

Round 2

Reviewer 1 Report

Comments and Suggestions for Authors

The authors addressed the concerns.

Author Response

The authors addressed the concerns.

Thank you for your feedback. We appreciate your time and are glad the revisions addressed your concerns.

Reviewer 2 Report

Comments and Suggestions for Authors

The authors have made changes in response to comments. The quality of the manuscript has improved significantly. It should be accepted.

Author Response

The authors have made changes in response to comments. The quality of the manuscript has improved significantly. It should be accepted.

Thank you for your feedback. We appreciate your time and are glad the revisions addressed your concerns.

Reviewer 3 Report

Comments and Suggestions for Authors

Review comments are attached

Comments on the Quality of English Language

Review comments are attached

Author Response

The revised form of the manuscript is improved over the originally submitted one. The corrections, however, introduced by the authors generated new problems that should not have been there in the first place. Therefore, certain points have been remarked upon that require the attention of the authors.

In the section “2.4.3. Effects on isolated rat brain microsomes”, the newly introduced statement “Noteworthy, compound 1 increased MDA production by 33%, 7 by 29%, 9 by 38%, 11 by 41%, 12 by 35%, 14 by 39%, and 15 by 40%, compared to the control.” should be replaced by “It is worth point out the fact that compound 1 increased MDA production by 33%, compound 7 by 29%, compound 9 by 38%, compound 11 by 41%, compound 12 by 35%, compound 14 by 39%, and compound 15 by 40%, compared to the control.” The authors should provide their input in fully written out statements, not in telegrams!

The corrections have been made.

In point 8 of the previous review process, the authors presented their view on the issue yet they provided no input to the ostensible observation made through the specific figure. Since they had previous seen differences, appropriately juxtaposed phenomena should be dioscussed in the manuscript and the specific location.

The necessary text is included.

A significant linguistic revision of the new version of the manuscript should be undertaken.

A full language revision of the text has been done.

Based on the aforementioned remarks, the manuscript should undergo corrections, after which it can be accepted.

Reviewer 4 Report

Comments and Suggestions for Authors

The revised version of the manuscript can be recommended for publication

Author Response

The revised version of the manuscript can be recommended for publication

Thank you for your feedback. We appreciate your time and are glad the revisions addressed your concerns.